# Joint Geometry–Appearance Human Reconstruction in a Unified Latent Space via Bridge Diffusion

## Abstract

Reconstructing both the geometry and appearance of a digital human from a single image remains highly challenging. Existing approaches typically decouple geometry and appearance, employing separate models for each, which limits their ability to reconstruct digital humans in a unified manner. In this paper, we propose **JGA-LBD**, which formulates human reconstruction as a bridge diffusion task in a unified latent space, yielding a joint latent representation that encodes both geometry and appearance. We address the challenge of human reconstruction from heterogeneous conditions, i.e., depth maps and SMPL models estimated from RGB images. Directly combining heterogeneous modalities introduces substantial training difficulties. To overcome this, we unify all conditions into 3D Gaussian representation and compress them into a unified latent space using a sparse variational autoencoder. All diffusion learning is then conducted within this unified latent space, which markedly reduces optimization complexity. Our setting strikingly lends itself to bridge diffusion: the depth map can be regarded as a partial observation of the target latent code, enabling the model to focus solely on inferring the missing components. Finally, a decoding module reconstructs geometry and renders novel-view images from the latent representation. Experiments demonstrate that JGA-LBD outperforms state-of-the-art methods in both geometry and appearance, and generates plausible results on in-the-wild images.

## 1 Introduction

Reconstructing high-fidelity digital humans from single-view RGB images is a fundamental problem in computer vision, with wide applications in virtual reality, gaming, autonomous driving, , *etc.*. Despite recent advances, achieving accurate reconstruction of both human geometry and appearance from a single image remains highly challenging, due to complex body shapes, diverse clothing, and severe self-occlusions.

Existing methods for digital human reconstruction can be broadly grouped into implicit function-based (Saito et al., 2019; 2020; Zhang et al., 2023c; 2024b; Ho et al., 2024), explicit point-based (Tang et al., 2025b; Han et al., 2023; Zhuang et al., 2025), and image-generation-based approaches (Zhang et al., 2025; Li et al., 2025a). Implicit function methods (Saito et al., 2019; 2020; Zhang et al., 2023c; 2024b; Ho et al., 2024) extract pixel-aligned features, features from parametric human models such as SMPL, or other cues, and use MLPs to learn occupancy fields or SDFs for surface reconstruction. While effective for geometry, they often fail to produce accurate appearance because query points in 3D space rarely have exact color supervision; the closest surface point is typically used as a proxy, causing the model to learn an approximation rather than ground truth colors. Explicit point-based methods (Tang et al., 2025b; Han et al., 2023; Zhuang et al., 2025) represent humans with point clouds derived from RGB images, often via estimated depth maps. These approaches can reconstruct detailed geometry, but typically ignore appearance, or build appearance models on top of pre-reconstructed geometry or parametric models like SMPL, resulting in multi-stage pipelines that may produce inconsistencies. Image-generation-based methods (Zhang et al., 2025; Li et al., 2025a) leverage large generative models to synthesize multi-view images from a single input view, and then reconstruct geometry using techniques such as continuous remeshing (Palfinger, 2022). While promising, they also require multiple steps and are sensitive to artifacts in the synthesized views.

Existing methods either struggle with appearance, focus solely on geometry or rely on complex, multi-stage pipelines. In summary, existing methods either struggle with appearance, focus solely on geometry, or rely on complex multi-stage pipelines. These limitations highlight two key requirements that remain unmet: first, accurate ground-truth of both geometry and appearance supervision is necessary; second, a single-stage method capable of jointly reconstructing geometry and appearance is required. Recently, 3D Gaussian representation (Kerbl et al., 2023) has achieved remarkable success in digital human modeling (Zhuang et al., 2025; Zhang et al., 2025; Qiu et al., 2025). As an explicit representation, it naturally encodes both geometry and appearance, thereby effectively addressing the need for reliable ground-truth supervision in joint reconstruction. However, the second requirement remains open: high-resolution modeling—typically involving over 100k Gaussians—poses a major challenge, namely how to efficiently process and generate such large-scale representations. A natural solution is to compress 3D Gaussians into a compact latent space and perform generative modeling there, leveraging diffusion models' strength in high-dimensional distribution learning. Yet, existing 3D diffusion approaches fall short: 3DShape2VecSet-based methods (Zhang et al., 2023a) only encode implicit fields and cannot capture appearance, while Trellis (Xiang et al., 2025) requires generating intermediate sparse structures before learning structured latents, preventing single-stage generation.

In this work, we present **JGA-LBD**, a bridge diffusion model that learns in a unified latent space and enables single-step reconstruction of high-resolution 3D Gaussians of digital humans. Specifically, we design a sparse VAE jointly trained with geometry and appearance supervision, which maps input 3D Gaussians into compact latent representations. To fully exploit the rich information embedded in images, we extract two complementary modalities—depth estimation and SMPL prediction—from the input. However, their inherent discrepancies make direct utilization challenging. To address this, we introduce a modality unification module that transforms both modalities into 3D Gaussian representations, which are subsequently compressed into the same latent space by the sparse VAE. This design ensures that all subsequent diffusion learning is carried out in a unified latent space, substantially reducing training complexity. Building on this unified latent design, we reveal that bridge diffusion offers an unexpectedly suitable framework for human reconstruction, since the depth-conditioned latent naturally corresponds to a partial observation of the target latent code. Rather than generating from noise, the bridge diffusion model only needs to complete the missing components, thereby significantly reducing the generative difficulty and improving the quality of the learned latent representations. Finally, the decoded 3D Gaussians from the latent code enables both geometry surface extraction and high-quality novel-view rendering via splatting-based rasterization. Extensive experiments on two benchmarks, together with evaluations on in-the-wild images, consistently demonstrate that JGA-LBD outperforms state-of-the-art methods in both quantitative accuracy and qualitative visual realism.

In summary, our contributions are:

- we design a sparse VAE that jointly compresses geometry and appearance of high-resolution 3D Gaussian representations into a compact latent code, overcoming prior methods that either focus solely on geometry or rely on additional sparse structural priors;
- we introduce a modality unification module that converts depth estimation and SMPL prediction into latent structural guidance through a sparse U-Net and an SMPL inpainter, ensuring consistent conditioning across heterogeneous modalities; and
- we adapt bridge latent diffusion to operate in the unified latent space, enabling efficient single-stage generation of complete latent codes, simultaneously modeling geometry and appearance.

## 2 RELATED WORK

**Diffusion Models** (Ho et al., 2020) have achieved remarkable success in generative modeling across diverse domains, including image synthesis, video generation, and audio processing. The core principle is to learn data distributions by gradually denoising Gaussian noise. While powerful, directly performing the diffusion process in pixel space is computationally expensive and often redundant. To address this, latent diffusion (Rombach et al., 2022) compresses the input into a compact latent space before applying the diffusion process, enabling efficient training while retaining high-quality generation. This paradigm has since become the standard for large-scale image and video diffusion models (Batifol et al., 2025; Peebles & Xie, 2023; Melnik et al., 2024).

A parallel line of work focuses on conditional diffusion, which aims to guide generation with auxiliary inputs. Early approaches such as classifier guidance and classifier-free guidance (Dhariwal & Nichol, 2021; Ho & Salimans, 2021) inject conditional signals during the sampling process. Later methods, such as ControlNet (Zhang et al., 2023b), extend this idea by introducing trainable networks that modulate intermediate features with external conditions, achieving fine-grained controllability. Despite their effectiveness, these methods still initialize the diffusion process from Gaussian noise, which may limit their ability to fully exploit structured priors. Bridge diffusion models (Zhou et al., 2024b; Li et al., 2023) address this limitation by replacing the Gaussian prior with a condition-driven source distribution, offering a more natural and efficient way to incorporate external structure. Besides, there are some optimization-based methods, e.g., TeCH (Huang et al., 2024), Human-SGD (AlBahar et al., 2023), WonderHuman (Wang et al., 2025b) and GeneMAN (Wang et al., 2025a)can also tackle this task with good generalization ability. PSHuman (Li et al., 2025a) and MagicMan He et al. (2024) use diffusion models to generate multi-view images and reconstruct the 3D human with the generated multi-view images. Human-GIF (Hu et al., 2025) formulates the human reconstruction task as a single-view conditioned human diffusion generation task. In this work, we build upon this idea and adopt bridge diffusion to leverage structural priors extracted from depth, which provide strong guidance for the generative process.

**3D Generative Models.** Generating 3D models is inherently more challenging than 2D image or video synthesis due to the diversity of 3D representations, which has led to two main research directions: multi-view based generation and direct 3D representation generation. Multi-view approaches first synthesize multiple 2D views and then reconstruct 3D content. For example, Zero123 (Liu et al., 2023) employs Stable Diffusion to generate multi-view images, after which a NeRF is optimized—following the SJC formulation (Wang et al., 2023)—to fit these synthesized views, and meshes are extracted via marching cubes from the learned density field. Leveraging the higher efficiency of 3D Gaussian Splatting (3DGS) compared to NeRF, methods such as DreamGaussian (Tang et al., 2024b) and LGM (Tang et al., 2024a) use image diffusion to produce multi-view images and subsequently fit 3DGS, though they often struggle to deliver high-resolution meshes. In contrast, direct 3D generation methods bypass multi-view supervision. DiffGS (Zhou et al., 2024a) encodes a 3DGS scene into a triplane latent and learns in latent space with DiT (Peebles & Xie, 2023); Crafts-Man3D (Li et al., 2025b) compacts shapes into vecsets (Zhang et al., 2023a) and trains DiT to learn an implicit field before extracting meshes at inference; and Trellis (Xiang et al., 2025) compresses 3DGS with sparse CNNs to support multiple downstream representations but requires an additional stage to provide geometric cues (sparse structure) and cannot unify geometry and appearance within a single latent. In contrast, our framework jointly compacts geometry and appearance into a unified latent representation and employs bridge diffusion to learn it in a single stage.

**Implicit-based 3D Human Reconstruction.** PIFu (Saito et al., 2019) is a pioneering work that reconstructs colored 3D humans using pixel-aligned features. Subsequent methods enhance implicit representations with additional cues: SiTH (Ho et al., 2024) generates a back-view image via ControlNet and uses a skinned mesh to resolve 3D ambiguity; GTA (Zhang et al., 2023c) introduces a ViT-based encoder–decoder to reconstruct clothed avatars with tri-plane features; and SIFU (Zhang et al., 2024b) leverages SMPL-X–guided cross-attention and a diffusion-based texture refinement pipeline to improve robustness in the wild. Despite these advances, implicit approaches lack ground-truth color supervision—appearance is approximated from the nearest surface point—limiting their ability to model high-fidelity textures.

**3DGS-based 3D Human Reconstruction.** Recently, 3DGS has emerged as a powerful explicit representation for human reconstruction. IDOL (Zhuang et al., 2025) leverages a large-scale dataset and a transformer-based predictor to reconstruct animatable Gaussian avatars efficiently. MultiGo (Zhang et al., 2025) introduces multi-level geometry learning with skeleton, joint, and wrinkle refinement, while LHM (Qiu et al., 2025) employs a multimodal transformer to preserve fine clothing and facial details. Chen et al. (2025) proposed a generate-then-refine pipeline and an HGM module to generate high-quality human 3D Gaussian attributes. HumanSplat (Pan et al., 2024) uses a video diffusion model for generating human 3D Gaussian attributes within a universal Transformer framework. These methods demonstrate the strength of 3DGS in capturing both geometry and appearance, though challenges remain in compact representation learning and efficient generative modeling.

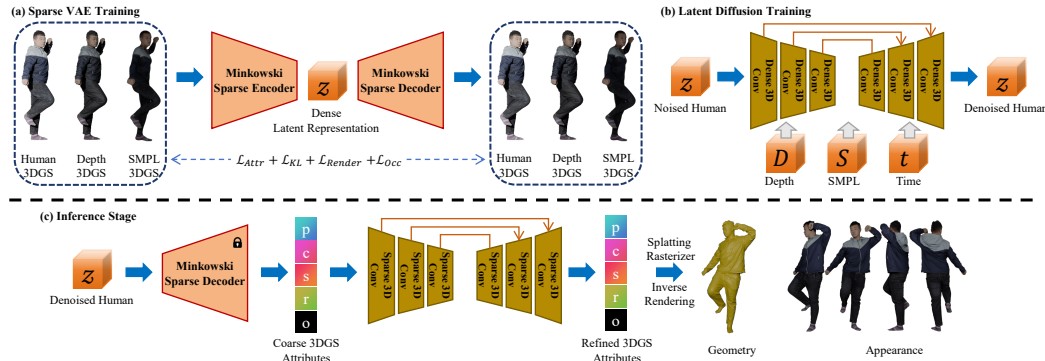

Figure 1: The pipeline of JGA-LBD. Given a single-view RGB image, depth and SMPL priors are converted into 3D Gaussians, which are compressed into latent codes by a sparse VAE. A bridge diffusion model generates latent codes conditioned on depth and SMPL priors, and the decoder refines them into a high-fidelity 3D Gaussian representation for surface reconstruction and novel-view rendering.

## 3 PROPOSED METHOD

JGA-LBD is a 3DGS-based method that reconstructs a 3D Gaussian representation $\mathcal{G} = \{G_1, ..., G_n\}$ from a single-view RGB image, where each element $G_i = \{p_i, c_i, s_i, r_i, o_i\}$ encodes its corresponding 3D Gaussian attributes such as voxel grid $p_i$, color $c_i$, scale $s_i$, rotation $r_i$ and opacity $o_i$. The resulting representation supports high-fidelity 3D surface reconstruction and novel-view appearance synthesis. To provide reliable supervision, we prepare ground-truth 3D Gaussians following the regularizations of (Tang et al., 2025a), with each scene containing about 130k∼150k Gaussians. As illustrated in Fig. 1, JGA-LBD consists of four key modules: (**i**) a modality unification module to unify the depth maps and SMPL vertices into the same sparse 3D Gaussian format; (**ii**) a sparse VAE that compresses both human 3D Gaussians and the converted conditions into latent codes in a unified latent space; (**iii**) a bridge diffusion model that learns the distribution of latent human 3D Gaussians conditioned on structural priors; and (**iv**) a decoder that transforms the denoised latent code back to a 3D Gaussian representation, followed by refinement to improve its fidelity before surface reconstruction and novel-view rendering. In what follows, we will detail each module.

### 3.1 MODALITY UNIFICATION MODULE

From a single RGB image, both a depth map and a corresponding SMPL model can be obtained. The depth map can be projected into a partial point cloud using camera parameters, while the SMPL model provides a complete geometric prior. Although both exist in 3D space, they belong to distinct modalities: the point cloud encodes $(x, y, z, r, g, b)$ values with appearance information, whereas the SMPL mesh contains only geometric vertices and faces. This discrepancy prevents them from being directly used as unified conditional inputs for supervision. To address this, both modalities are first transformed into a consistent 3D Gaussian representation, achieving modality unification.

For the partial point cloud, we first perform nearest-neighbor search to associate each point with its closest Gaussian in the target human 3D Gaussians, using these attributes as ground truth for supervision. A sparse U-Net based on Minkowski Engine (Choy et al., 2019) is then trained to map $(r, g, b)$ values of the partial point cloud to 3D Gaussian attributes. For the SMPL mesh, we first project it onto the image plane, where only visible vertices receive color information, leaving occluded vertices uncolored. This partially observed mesh is then passed through another sparse Minkowski U-Net to predict complete 3D Gaussian attributes for all vertices, effectively generating 3D Gaussian representation. Note that the resulting colors are coarse and not intended as precise appearance supervision, this process primarily ensures that SMPL provides global structural guidance in a unified 3DGS format. By transforming heterogeneous modalities into 3D Gaussians, we obtain consistent and complementary conditional inputs for the diffusion process.

## 3.2 JOINT GEOMETRY-APPEARANCE COMPRESSION VAE

Voxel is a common 3D representation that is compatible with standard CNNs. However, accurately representing a 3D object typically requires very high-resolution voxel grids (at least $512^3$), which is infeasible for training due to excessive GPU memory requirements. Inspired by latent diffusion Rombach et al. (2022), we employ a sparse VAE to compress 3D Gaussians into a compact latent representation, enabling efficient modeling with standard CNNs in a unified latent space.

Given the ground-truth human 3D Gaussian attributes $\mathcal{G}$, converted depth 3D Gaussian attributes $\mathcal{D}$, and SMPL 3D Gaussian attributes $\mathcal{S}$, our goal is to encode them into a unified latent representation that jointly captures both geometry and appearance. Specifically, we build the sparse VAE with Minkowski Engine (Choy et al., 2019). The encoder $E(\cdot)$ of the sparse VAE consists of several resnet blocks, the output $z$ of encoder $E(\cdot)$ serves as the ground truth for a diffusion model. To avoid learning a high-variance latent space, we impose a slight KL-penalty $\mathcal{L}_{\text{KL}}$ to $z$ to make it learn a latent with a standard normal distribution. The decoder $D(\cdot)$ is a key module in the sparse VAE, as it should decode the denoised $z$ of the diffusion model independently without any sparse structure cues like Trellis (Xiang et al., 2025). Hence, we adopt the generative sparse transpose convolution layers to build the decoder $D(\cdot)$, which enables generating of new coordinates that do not need the cache coordinates from the encoder as in standard sparse transpose convolutions. It starts from $z$ and proceeds by progressively pruning excessive voxels with the occupancy loss $\mathcal{L}_{\text{Occ}}$, and finally reaching the resolution of $\mathcal{G}$ after several layers. We use MSE loss to supervise the reconstruction of 3D Gaussian attributes, however, the predicted voxel grid and the ground-truth voxel grid are not strictly aligned and we cannot directly apply the MSE loss on the sparse tensors. Converting sparse tensors into dense form introduces a vast number of non-active voxels (e.g., only about 130k∼150k active voxels out of $512^3$), which seriously dilutes gradients and hinders effective learning. Therefore, we compute the MSE loss only on the intersection of active voxels between the prediction and the ground truth:

$$\mathcal{L}_{\text{Attr}} = \frac{1}{|\mathcal{I}|} \sum_{i \in \mathcal{I}} \|\mathbf{a}_p(i) - \mathbf{a}_g(i)\|_2^2, \tag{1}$$

where $\mathcal{I}$ denotes the intersection of active voxel indices, and $\mathbf{a}_p(i)$, $\mathbf{a}_g(i)$ are the predicted and gt 3D Gaussian attributes (i.e., $p, c, s, r, o$) at location $i$. However, supervising only on the intersection inevitably leaves certain regions unsupervised. To address this limitation, we introduce a loss by rendering the predicted 3DGS into 2D images and enforcing consistency with the ground-truth images. The rendering loss combines the L1 loss, SSIM loss and LPIPS loss:

$$\mathcal{L}_{\text{Render}} = \lambda_1 \|I_p - I_g\|_1 + \lambda_2 \big(1 - \text{SSIM}(I_p, I_g)\big) + \lambda_3 \text{LPIPS}(I_p, I_g), \tag{2}$$

where $I_p$ and $I_g$ denote the rendered and gt images, respectively, and $\lambda_1, \lambda_2, \lambda_3$ are balancing weights. The overall training objective of the sparse VAE is:

$$\mathcal{L}_{\text{VAE}} = \lambda_4 \mathcal{L}_{\text{KL}} + \lambda_5 \mathcal{L}_{\text{Occ}} + \lambda_6 \mathcal{L}_{\text{Attr}} + \lambda_7 \mathcal{L}_{\text{Render}}, \tag{3}$$

where $\lambda_4, \lambda_5, \lambda_6, \lambda_7$ are balancing weights. The encoded results of $\mathcal{G}, \mathcal{D}$ and $\mathcal{S}$ are converted to dense latent representations for the diffusion training, denoted as $\mathcal{G}_L, \mathcal{D}_L$ and $\mathcal{S}_L$ respectively.

## 3.3 BRIDGE DIFFUSION IN UNIFIED LATENT SPACE

Diffusion models are typically designed to transport data distributions into a standard Gaussian prior. However, in our setting, the depth-derived latent code $\mathcal{D}_L$ can be regarded as a structural subset of the full human Gaussian representation. Thus, instead of relying on diffusion models, we adopt the more powerful bridge diffusion model (Zhou et al., 2024b), which learns a transport path between two arbitrary distributions. Specifically, the goal is to translate from the structural prior distribution $p_{\mathcal{D}_L}$ to the target distribution $p_{\mathcal{G}_L}$, while being conditioned on the SMPL prior $\mathcal{S}_L$.

Formally, a bridge diffusion process is represented by a sequence of time-indexed variables $\{x_t\}_{t=0}^T$. Using Doob's $h$-transform (Doob & Doob, 1984), the conditional stochastic bridge can be expressed as:

$$dx_t = f(x_t, t \mid \mathcal{S}_L)\, dt + g(t)^2\, h(x_t, t, y, T \mid \mathcal{S}_L)\, dt + g(t)\, dw_t, \tag{4}$$

where $f(x_t, t \mid \mathcal{S}_L)$ is the drift term and $g(t)$ is the diffusion coeff, $x_0 \sim p_{\mathcal{G}_L}(x \mid \mathcal{S}_L)$, $x_T = y$, and $y \sim p_{\mathcal{D}_L}$. The term $h(x, t, y, T \mid \mathcal{S}_L) = \nabla_x \log p(x_T = y \mid x_t = x, \mathcal{S}_L)$ denotes the

Table 1: Quantitative comparisons of different methods on 2K2K and CustomHuman. The best results are highlighted in **bold**. ↑: the higher the better. ↓: the lower the better.

| Metric Method | 2K2K | | | | | | CustomHuman | | | | | |
|---|---|---|---|---|---|---|---|---|---|---|---|---|
| | PSNR↑ | SSIM↑ | LPIPS↓ | CD↓ | P2S↓ | Normal↓ | PSNR↑ | SSIM↑ | LPIPS↓ | CD↓ | P2S↓ | Normal↓ |
| GTA (NeurIPS 23) | 24.15 | 0.921 | 0.080 | 1.156 | 1.114 | 2.127 | 28.86 | 0.920 | 0.088 | 1.249 | 1.123 | 2.552 |
| SIFU (CVPR 24) | 23.47 | 0.910 | 0.088 | 1.154 | 1.135 | 2.180 | 29.62 | 0.928 | 0.092 | 1.365 | 1.205 | 2.696 |
| SiTH (CVPR 24) | 24.30 | 0.920 | 0.076 | 0.891 | 0.944 | 2.019 | 26.47 | 0.911 | 0.095 | 2.244 | 2.367 | 3.365 |
| PSHuman (CVPR 25) | 24.72 | 0.917 | 0.067 | 0.575 | 0.608 | 1.440 | 30.26 | 0.931 | 0.082 | 1.055 | 1.146 | 1.899 |
| IDOL (CVPR 25) | 27.18 | 0.929 | 0.076 | 1.138 | 1.138 | 2.454 | 31.02 | 0.934 | 0.076 | 1.119 | 1.188 | 2.416 |
| Human3Diffusion (NeurIPS 24) | 29.05 | 0.942 | 0.062 | 0.503 | **0.415** | 1.429 | **33.75** | 0.952 | 0.067 | 0.809 | 0.768 | 1.755 |
| MultiGO (CVPR 25) | 28.80 | 0.939 | 0.059 | 0.636 | 0.655 | 1.474 | 31.72 | 0.934 | 0.075 | 1.750 | 1.809 | 2.440 |
| Trellis (CVPR 25) | 25.47 | 0.927 | 0.069 | 0.771 | 0.743 | 1.929 | 31.33 | 0.934 | 0.069 | 1.202 | 1.219 | 2.370 |
| JGA-LBD | **30.16** | **0.946** | **0.055** | **0.489** | 0.507 | **1.202** | 33.44 | **0.957** | **0.061** | **0.674** | **0.670** | **1.469** |

drift adjustment introduced by the $h$-transform to ensure that the process interpolates between the endpoints. Reversing this bridge process yields the conditional reverse SDE

$$dx_t = \Big[ f(x_t, t \mid \mathcal{S}_L) - g(t)^2 \big( U_\theta(x_t, t, y, T \mid \mathcal{S}_L) - h(x_t, t, y, T \mid \mathcal{S}_L) \big) \Big] dt + g(t) \, d\bar{w}_t, \quad (5)$$

and the associated probability flow ODE

$$dx_t = \Big[ f(x_t, t \mid \mathcal{S}_L) - g(t)^2 \big( \tfrac{1}{2} U_\theta(x_t, t, y, T \mid \mathcal{S}_L) - h(x_t, t, y, T \mid \mathcal{S}_L) \big) \Big] dt, \quad (6)$$

where $U_\theta$ denotes the neural network with parameters $\theta$ approximation of the bridge score function. To learn this score function, we adopt denoising bridge score matching, which minimizes the discrepancy between the predicted score and the closed-form conditional score of the Gaussian bridge:

$$L(\theta) = \mathbb{E}_{x_t, x_0, x_T, t} \Big[ w(t) \left\| U_\theta(x_t, x_T, t \mid \mathcal{S}_L) - \nabla_{x_t} \log q(x_t \mid x_0, x_T, \mathcal{S}_L) \right\|^2 \Big], \quad (7)$$

where $w(t)$ denotes a time-dependent weighting function that adjusts the relative importance of different diffusion steps during training.

***Remark.*** We augment each grid of the dense latent representation with an occupancy value $\{0, 1\}$, allowing the bridge diffusion model to jointly learn both the latent features and their occupancy. During inference, the dense latent representation is converted into a sparse latent representation by retaining only the grids with predicted occupancy greater than 0.5.

## 3.4 DECODE MODULE

We compress the original sparse 3D Gaussian attributes of size $(512^3, 20)$ into a latent representation of size $(64^3, 4)$ using a sparse VAE, significantly reducing GPU memory consumption and enabling feasible training. However, this aggressive compression inevitably leads to information loss, which is further exacerbated after the diffusion generation process. To alleviate this, following recent advances in large-scale 3D generative models (Xiang et al., 2025; Ren et al., 2024; Li et al., 2025b), we append a Minkowski Engine–based U-Net after the VAE decoder to refine the outputs of the diffusion model.

**Mesh Extraction.** For each reconstructed 3D Gaussians, we recover the human surface using its position attributes. Vertex normals are estimated via WNNC (Lin et al., 2024), and the surface is reconstructed with screened Poisson (Kazhdan & Hoppe, 2013). To further enhance geometric fidelity, the reconstructed surface is refined using depth supervision: the surface is rendered into a depth map with PyTorch3D, and an L1 loss is computed against the predicted depth map.

**Novel View Synthesis.** For novel-view rendering, we adopt the standard 3D gaussian splatting pipeline, where the refined 3D Gaussians is rendered from arbitrary viewpoints using the corresponding camera parameters.

## 4 EXPERIMENTS

**Datasets.** We conduct experiments on Thuman2.1 (Yu et al., 2021), 2K2K (Han et al., 2023) and CustomHuman (Ho et al., 2023). Specifically, 1600 scans from Thuman2.1 are used as ground truth to prepare the ground truth 3D Gaussian attributes, which are used for training the sparse VAE and the bridge latent diffusion model. For evaluation, we use 25 scans from 2K2K and 40 scans

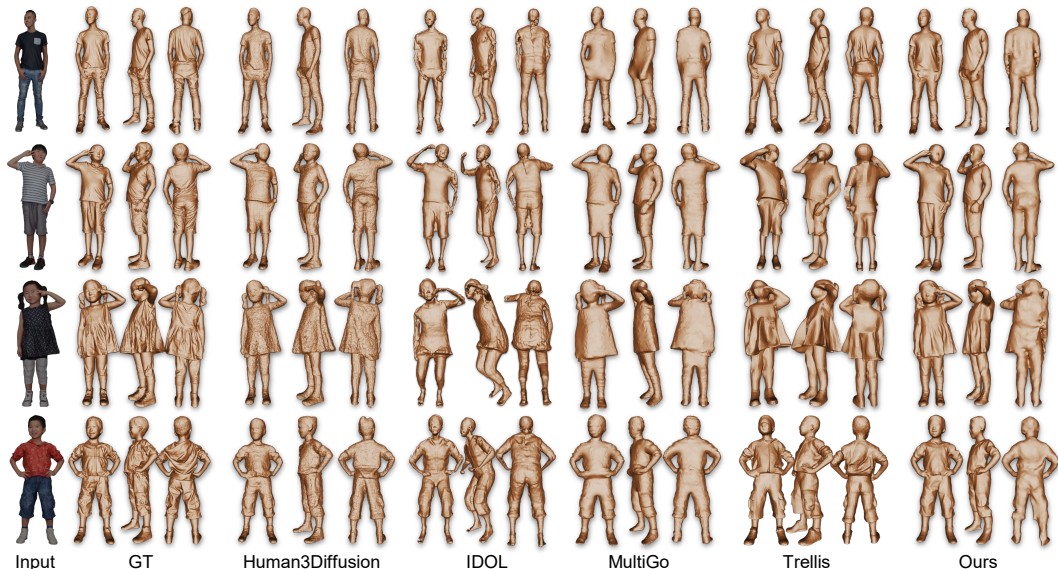

Figure 2: Geometry comparisons of our method against 3DGS-based methods, *i.e.*, Human3Diffusion, IDOL, MultiGo and Trellis. 🔍 Zoom in for details.

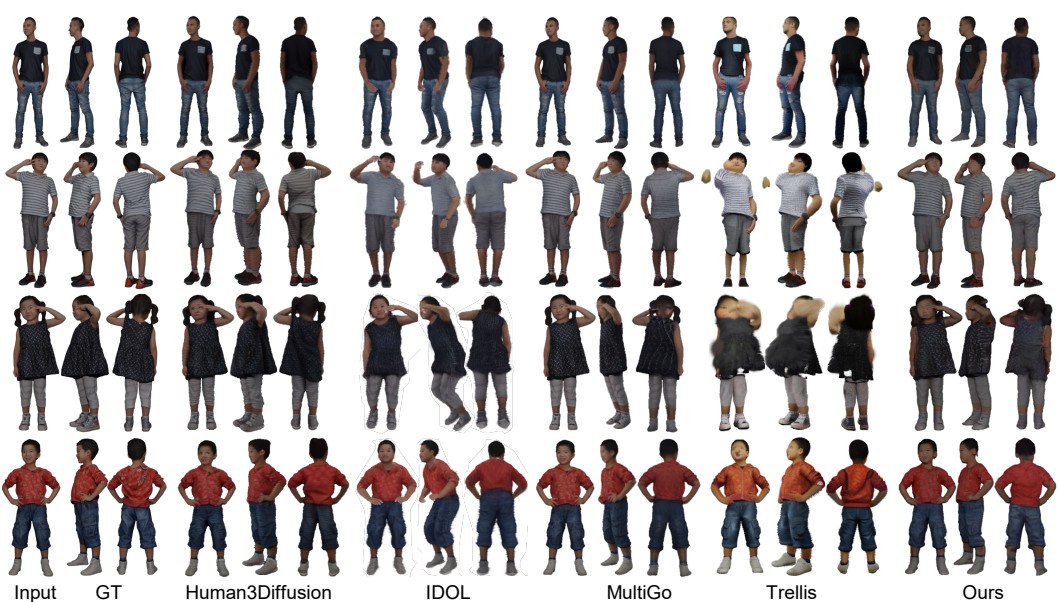

Figure 3: Appearance comparisons of our method against 3DGS-based methods, *i.e.*, Human3Diffusion, IDOL, MultiGo and Trellis. 🔍 Zoom in for details.

from CustomHuman. In addition, we assess the generalization ability of JGA-LBD on in-the-wild images collected from the Internet.

**Evaluation metrics.** All 3DGS and mesh outputs are normalized to the cube $(-1, 1)$. For appearance reconstruction, we report peak signal-to-noise ratio (PSNR), structural similarity index (SSIM), and learned perceptual image patch similarity (LPIPS). For geometry reconstruction, we evaluate Chamfer distance (CD), point-to-surface distance (P2S), and normal error.

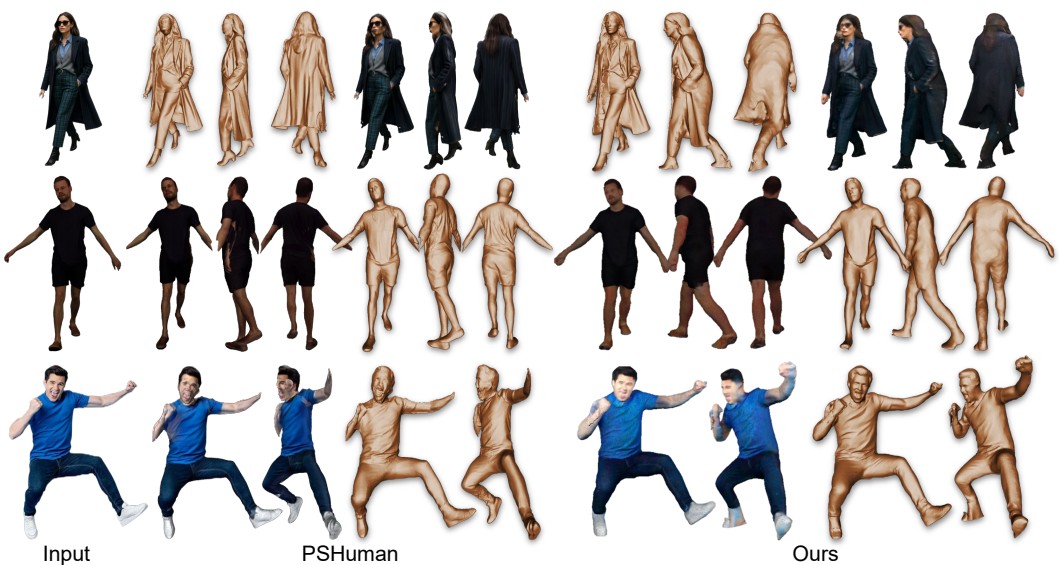

Figure 4: Geometry and appearance comparisons of our method against PSHuman. 🔍 Zoom in for details.

## 4.1 COMPARISON WITH STATE-OF-THE-ART METHODS

We mainly compare our JGA-LBD with three representative 3DGS-based approaches. IDOL (Zhuang et al., 2025) leverages an explicit SMPL model as a geometry prior to guide 3D Gaussians generation. MultiGo (Zhang et al., 2025) generates a complete 3D Gaussian scene using a large model, while encoding SMPL as Fourier features to provide structural guidance. Trellis (Xiang et al., 2025) learns a compact latent space that jointly encodes both geometry and appearance for structured generative modeling with multiple stages. We further compare JGA-LBD with several recent mesh-based methods (Ho et al., 2024; Zhang et al., 2023c; 2024b; Li et al., 2025a). As shown in Table 1, JGA-LBD achieves the best performance across all metrics on both benchmark datasets. **Geometry Comparison.** Human3Diffusion (Xue et al., 2024) produces noticeably noisy geometry and fails to preserve fine frontal wrinkle details. IDOL (Zhuang et al., 2025) heavily relies on the SMPL model without any refinement, and therefore, as shown in Figure 2, it often produces incorrect poses. Moreover, due to the strong regularization imposed by SMPL, it fails to handle loose clothing such as dresses (see the third case). Although MultiGo (Zhang et al., 2025) employs a wrinkle refinement network, its geometric reconstruction still lacks fine details. Moreover, as observed in the third and fourth cases in Figure 2, the reconstructed bodies exhibit a forward-leaning tendency. This indicates that, although MultiGo avoids the pose inaccuracies introduced by directly using SMPL, its 2D diffusion model is insufficient to correct pose errors in 3D space. Trellis (Xiang et al., 2025) suffers from low mesh resolution, which severely limits the reconstruction of fine details. In addition, the reconstructed poses are often inaccurate, with head rotations consistently misaligned with the input across all cases. In contrast, our JGA-LBD is able to reconstruct fine geometric details and handle loose clothing, while maintaining accurate overall human poses.
**Appearance Comparison.** Human3Diffusion (Xue et al., 2024) shows many jagged artifacts in the rendered images, and it cannot recover texture details accurately. IDOL (Zhuang et al., 2025) suffers from severe misalignment caused by wrong SMPL poses. As shown in the Figure. 3, it can only capture relatively simple color patterns and fails to represent fine-grained textures such as stripes in the second case. In addition, noticeable jagged artifacts can be observed along the edges. MultiGo (Zhang et al., 2025) performs well on the front side, but its back-side reconstructions remain poor. For example, in the second case it fails to recover the stripe patterns, and in the third case the back of the head incorrectly contains facial details instead of black hair. Beyond its failure to reconstruct fine details such as stripes, Trellis (Xiang et al., 2025) also suffers from severe geometry–appearance inconsistencies. For instance, in the second and third cases, the reconstructed arms are noticeably inconsistent with those shown in Figure. 2. In contrast, our JGA-LBD not only reconstructs fine details such as stripes and back-side wrinkles, but also maintains geometric consistency with the

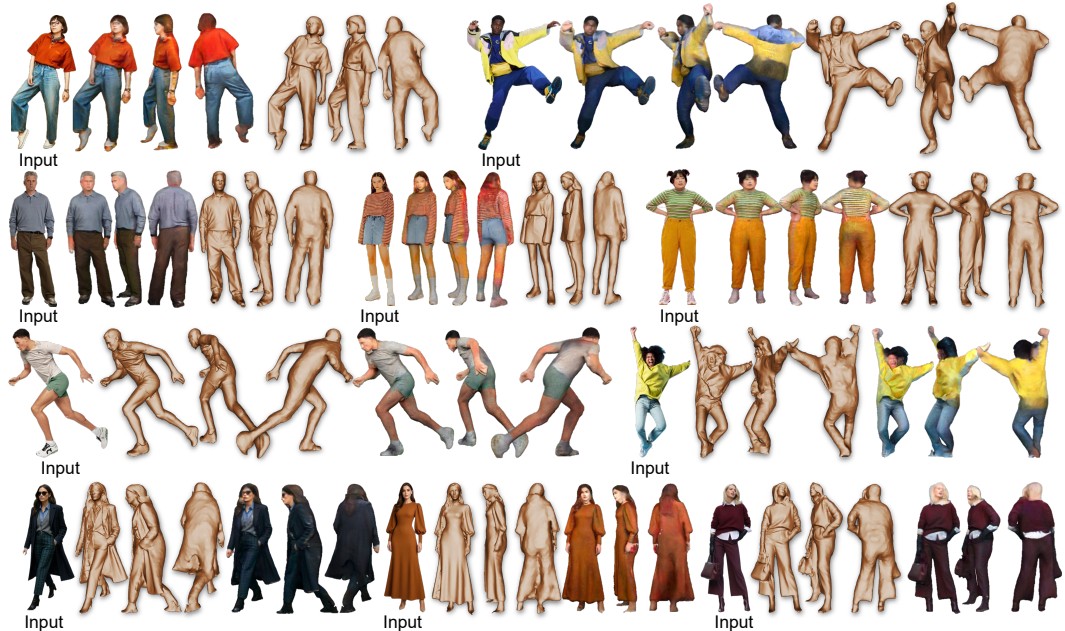

Figure 5: The reconstructed results of our JGA-LBD on in-the-wild images. 🔍 Zoom in for details.

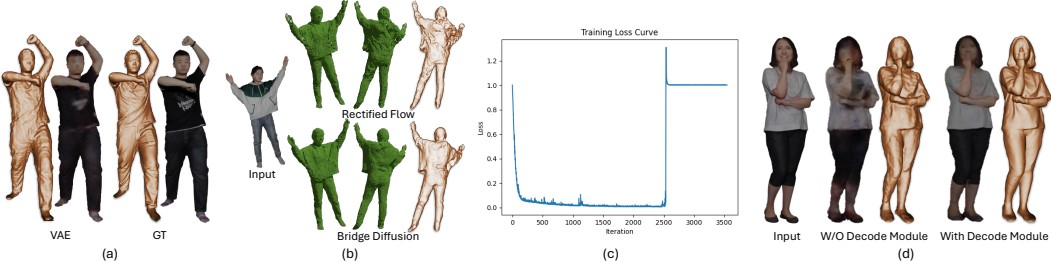

Figure 6: Ablation visual results. (a). The visualization comparison between VAE results and GT. (b). Comparison between rectified flow and bridge diffusion. (c). Training loss curve of image and SMPL feature level supervision. (d). Comparison between visual results without and with the decode module. 🔍 Zoom in for details.

reference in Figure. 2.

**Comparison with PSHuman (Li et al., 2025a).** We also compare our JGA-LBD with the recently popular PSHuman. As shown in Fig. 4, PSHuman frequently suffers from missing body parts. PSHuman also struggles with handling the relative positions of the legs in the first example. In the second example, it also fails to preserve facial details, producing noticeably degraded results. In contrast, our method effectively avoids these issues.

Overall, both quantitative metrics and qualitative comparisons demonstrate that our JGA-LBD consistently outperforms state-of-the-art methods. We further conducted experiments on in-the-wild images with challenging poses, as shown in Fig. 5, where JGA-LBD successfully reconstructs plausible appearances and detailed 3D surfaces. More visual results are shown in Fig. A1 in Appendix.

## 4.2 ABLATION STUDIES

**Visual Results of Sparse VAE.** To evaluate the effectiveness of the sparse VAE, we conduct experiments on reconstructing human 3D Gaussian representations directly from the compressed latent space. Specifically, the input 3D Gaussians are compressed from $(512^3, 20)$ into a latent tensor of size $(64^3, 4)$, and subsequently decoded to recover 3D Gaussian attributes. As shown in Fig. 6 (a), the sparse VAE is able to preserve the overall geometry and coarse appearance of the human

Table 2: Ablation studies on 2K2K.

| Metric / Method | PSNR↑ | SSIM↑ | LPIPS↓ | CD↓ | P2S↓ | Normal↓ |
|---|---|---|---|---|---|---|
| Rectified Flow | 28.32 | 0.931 | 0.074 | 0.568 | 0.535 | 1.436 |
| Feature Condition | Fail | Fail | Fail | Fail | Fail | Fail |
| w/o Decode Module | 27.68 | 0.931 | 0.073 | 0.498 | **0.497** | 1.287 |
| EcoDepth + PIXIE | 30.01 | 0.944 | 0.059 | 0.536 | 0.544 | 1.288 |
| DepthAnything + PyMAF | 29.51 | 0.943 | 0.059 | 0.540 | 0.543 | 1.367 |
| Full Model | **30.16** | **0.946** | **0.055** | **0.489** | 0.507 | **1.202** |

body, demonstrating that the latent space effectively encodes both structural and visual information. However, fine-grained details such as sharp geometric boundaries and high-frequency textures are noticeably degraded due to the high compression ratio. This observation motivates the introduction of a decode module after the VAE decoder to enhance reconstruction fidelity.

**Effectiveness of Bridge Diffusion.** We further compare the bridge diffusion employed in our work with the popular rectified flow method (Liu et al., 2022), the quantitative results are shown in Table 2. The visual results in Fig. 6 (b) show that rectified flow tends to generate 3D Gaussians with many holes, and the reconstructed back surfaces are heavily corrupted by noise. This observation highlights the advantage of bridge diffusion: since the starting depth is already part of the complete 3D Gaussians, bridge diffusion does not need to allocate excessive capacity to the visible front side but instead focuses on learning the missing regions. As a result, our strategy of adopting bridge diffusion achieves superior reconstruction quality.

**Necessity of Structural Prior.** We further validate the necessity of introducing structural priors. Following GaussianCube (Zhang et al., 2024a), we extract image features using DINOv2 (Oquab et al., 2023) and SMPL features using Point-M2AE (Zhang et al., 2022), and employ both features to supervise the training process of bridge diffusion model. However, the training often fails to converge: as shown in Fig. 6 (c), the loss decreases at the beginning but suddenly rises after several epochs, eventually leading to divergence. We attribute this to the modality gap between images and SMPL, which makes it difficult for the model to learn meaningful representations when directly using such heterogeneous features. This observation underscores the importance of our unified latent space, where features from different modalities are mapped into a shared representation, substantially reducing training difficulty and improving stability.

**Necessity of Decode Module.** Due to the aggressive compression ratio of the VAE, inevitable information loss is introduced. In addition, the diffusion model itself cannot achieve perfectly error-free reconstruction, and such residual errors further amplify the loss caused by compression, making it difficult to capture and recover high-frequency details. To address this issue, we introduce a decoding module after the diffusion model to refine and enhance the generated results. As shown in Table 2, the decode module brings significant improvement on both appearance and geometry metrics. Fig. 6 (d), we can also observe enhanced details in both appearance and geometry. Moreover, it is worth noting that even without the additional decode module, the results already achieve the best performance in terms of geometry and deliver appearance reconstruction that remains competitive with state-of-the-art methods. To quantify the effect of different depth estimation methods, we compare two depth estimators—Depth Anything V2 (Yang et al., 2024) (RMSE 0.014 on 2K2K) and EcoDepth (Patni et al., 2024) (0.016)—and observe that better depth leads to consistently better reconstruction performance (Table 2). We also evaluate different SMPL regressors (i.e., PIXIE (Feng et al., 2021) vs. PyMAF (Zhang et al., 2021)) and find that they produce comparable results with minor variations.

## 5 CONCLUSION

In this work, we have presented JGA-LBD, a framework that reconstructs both geometry and appearance of a human in a single generation step. Experimental results demonstrate that our method achieves superior performance in both geometry and appearance reconstruction compared to state-of-the-art methods. Unlike existing methods that decouple geometry and appearance, JGA-LBD performs joint modeling, thereby ensuring better consistency between geometry and appearance.

In future work, we will exploit a more powerful sparse VAE capable of capturing high-frequency details and explore diffusion architectures that eliminate the need for an additional refinement decoder, further improving both efficiency and reconstruction quality.

STATEMENT

ETHICS STATEMENT

We adhere to the IClR Code of Ethics in this research, The datasets used are publicly available with no inclusion of private, sensitive, or proprietary data involving human subjects.

REPRODUCIBILITY STATEMENT

Our work prioritizes reproducibility. All code for data pre-processing, model training, and evaluation will be made publicly available. The datasets used are all publicly accessible, and we have cited their corresponding literature in the paper, The experiments were run on a server with an Intel Xeon 4309Y CPU and 4 NVIDIA RTX A6000 GPU, using PyTorch 2.2, CUDA 12.

STATEMENT ON AI USE

We used an LLM, i.e., ChatGPT, solely for grammar polishing of the manuscript. All LLM outputs were manually verified for accuracy, and no content was directly adopted without validation. The authors bear full responsibility for all content.

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

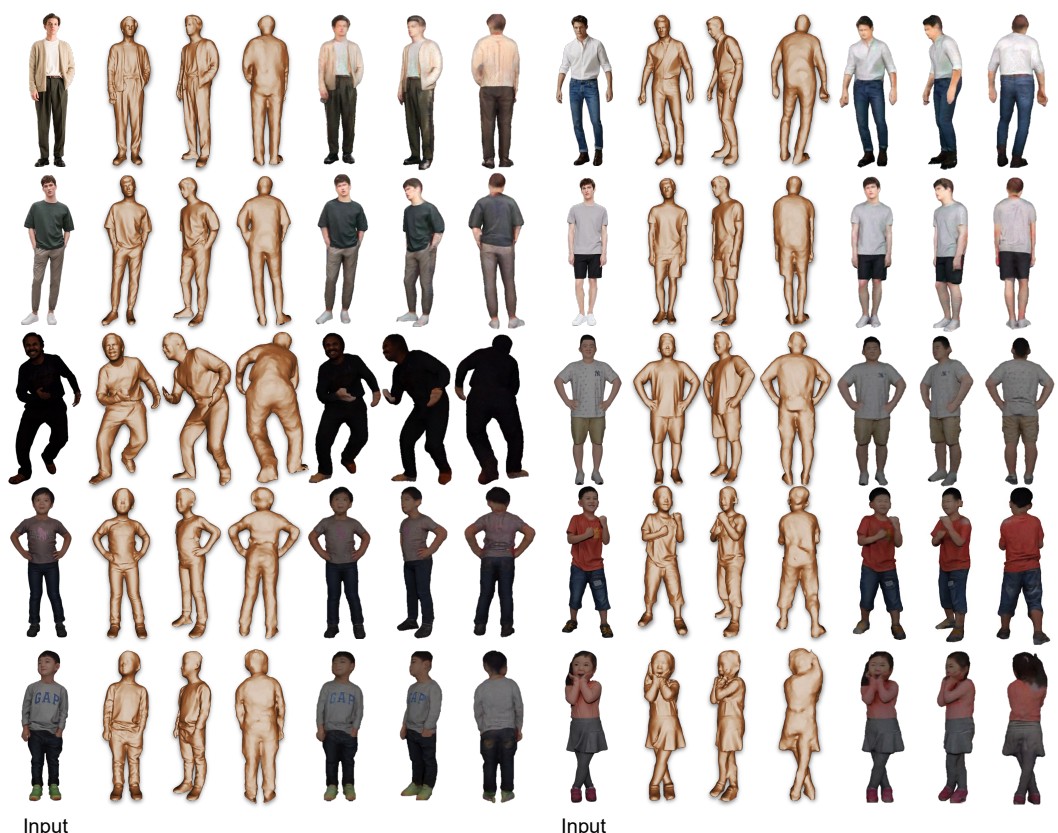

Figure A1: More visual results. 🔍 Zoom in for details.

# A APPENDIX

## A.1 IMPLEMENTATION DETAILS.

The sparse VAE was trained using Minkowski Engine 0.5.4 with a batch size of 8 for 200,000 iterations. The Adam optimizer was employed with a learning rate of 0.00035. The loss weights $\lambda_1 \sim \lambda_7$ were set as 0.8, 0.2, 0.1, $5 \times 10^{-7}$, 1, 1, 1, respectively. (To improve training stability, $\mathcal{L}_{\text{Attr}}$ and $\mathcal{L}_{\text{Render}}$ were introduced only after 10,000 iterations.) For bridge diffusion, both training and inference followed the original DDBM setting (Zhou et al., 2024b), where the parameters CHURN_STEP_RATIO and GUIDANCE were set to 0.1 and 1, respectively. The batch size was set to 16, and the number of training iterations was 100,000. The Adam optimizer was employed with a learning rate of 0.00035. Depth Anything V2 (Yang et al., 2024) was adopted as the backbone for depth estimation and PIXIE (Feng et al., 2021) was selected to predict the SMPL models. The sparse 3D UNet was trained with a batch size of 8 for 40,000 iterations. The Adam optimizer was employed with a learning rate of 0.00035. All training and testing were conducted on a server equipped with four NVIDIA A6000 GPUs.

The parameter counts of the sparse VAE, bridge-diffusion 3D U-Net, and sparse U-Net are 9.04M, 229.76M, and 36.99M, respectively. The training times for these models are 2 days, 4 days, and 1 day, respectively. The GPU memory cost for these models are 40 GB, 4×45 GB, and 30 GB, respectively. The inference time for a single sample is approximately 2 minutes.

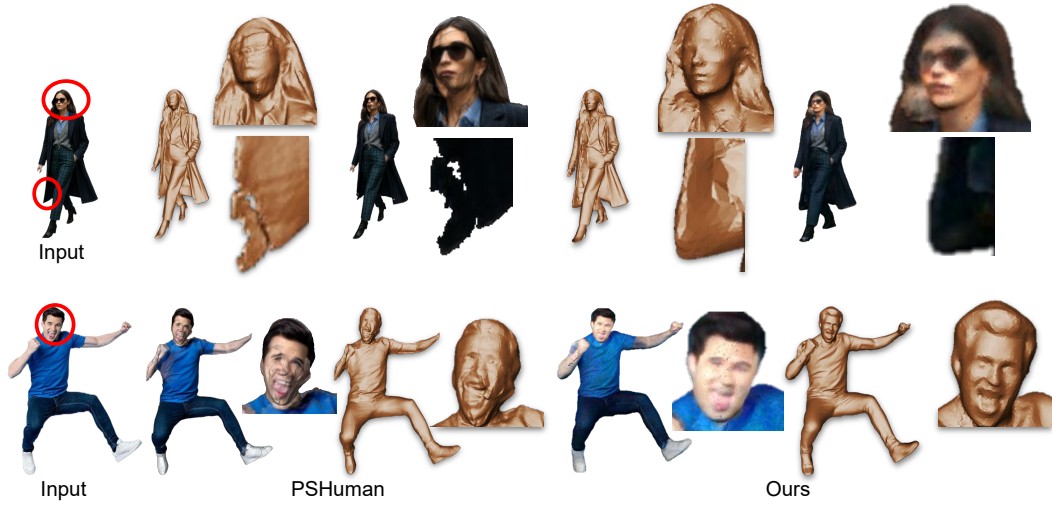

Figure A2: Comparisons of geometric and appearance details between our method and PSHuman. 🔍 Zoom in for details.

