# OpenReview forum: "Joint Geometry–Appearance Human Reconstruction in a Unified Latent Space via Bridge Diffusion"
_ICLR.cc/2026/Conference — ICLR 2026 Conference Withdrawn Submission_

### Official Review · Reviewer_gV1m · 2025-10-27

**Soundness:** 2
**Presentation:** 2
**Contribution:** 2
**Rating:** 2
**Confidence:** 4

**Summary:**

This paper proposes a method to unify heterogeneous modalities, such as depth maps and SMPL models, into 3D Gaussian representations. These modalities are jointly modeled in a unified latent space via a Sparse Variational Autoencoder (Sparse VAE), and a diffusion model is subsequently employed to generate complete latent codes.

- Datasets: The model is trained on Thuman2.1. Evaluation is conducted on the 2K2K (25 scans) and CustomHuman (40 scans) benchmarks. Furthermore, the model's generalization capabilities are validated on in-the-wild images.

- Metrics: Evaluation metrics include PSNR, SSIM, and LPIPS for texture quality, alongside Chamfer Distance (CD) and normal error for geometric accuracy.

**Strengths:**

1. The reconstruction of 3D Gaussian Splatting (3DGS) or mesh representations from a single image is a critical problem with numerous downstream applications.

2. The proposed method facilitates end-to-end training, bypassing multi-stage synthesis pipelines (e.g., multi-view image generation followed by geometric reconstruction). This design potentially enhances inference efficiency compared to conventional multi-step approaches. The paper presents a framework that jointly leverages depth priors (from DepthAnything), diffusion priors, and SMPL priors.

**Weaknesses:**

1. The integration of geometric priors into the diffusion process to guide generation is a relatively established technique, as seen in works like DiffSplat and Hunyuan3D. Furthermore, leveraging an SMPL prior as a structural constraint has been demonstrated in several existing methods (e.g., SiFU, SiTH), which calls the novelty of this specific contribution into question.

2. The paper provides insufficient visualization results, making it difficult to assess the 3D consistency of the proposed method. Based on prior work in human avatar generation, a minimum of `15-20 video results` are typically required to substantiate generalization claims. Comparative video analysis against other methods is also strongly recommended. The presented qualitative results are unconvincing. Specifically, Fig. 2 and Fig. 3 exhibit significant loss of detail in facial regions and clothing. Moreover, the model performs poorly in challenging scenarios, such as the loose-fitting dress depicted in the third row. This apparent qualitative failure contrasts sharply with the reported quantitative metrics (e.g., `PSNR 29.91`, `SSIM 0.943`), which are surprisingly high, approaching the performance ceiling typically associated with multi-view reconstruction.

3. A fundamental representational trade-off exists: mesh-based methods typically excel in geometric fidelity (e.g., sharp contours, fewer artifacts/floaters), whereas 3DGS representations generally offer superior texture modeling (e.g., facial details, complex clothing). Achieving state-of-the-art (SOTA) performance in both geometry and texture simultaneously within a single representation is non-trivial. It is recommended that the authors align their evaluation benchmark with established standards in existing literature.

4. The paper omits comparisons with several critical and highly relevant baselines in human avatar generation, notably HumanSplat [1], Human3Diffusion [2], and PSHuman [3]. Empirically, these human-centric models demonstrate superior texture fidelity and overall quality compared to general object generation models, making them essential points of comparison.

5. Given that the method jointly utilizes multiple priors (`DepthAnything V2`, a `diffusion prior`, and an `SMPL prior`), a comprehensive ablation study is required. This study must meticulously dissect the contribution of each component to quantify its relative importance and impact on the final performance.

6. `Critical implementation details are omitted`. For example, while citing relevant works [e.g., 4, 5], the authors do not specify the exact architecture of the diffusion model employed as a prior, its parameter count, or the computational runtime.


If the authors can address the concerns, I am willing to raise my score.


 [1] Human-3Diffusion: Realistic Avatar Creation via Explicit 3D Consistent Diffusion Models, NeurIPS 2024

[2] HumanSplat: Generalizable Single-Image Human Gaussian Splatting with Structure Priors. NeurIPS 2024

 [3] PSHuman: Photorealistic Single-image 3D Human Reconstruction using Cross-Scale Multiview Diffusion and Explicit Remeshing, CVPR 2025

[4] Zhou et al. Denoising diffusion bridge models. ICLR, 2024b

[5] xiang et al. Structured 3d latents for scalable and versatile 3d generation, CVPR 2025

**Questions:**

- What is the exact architecture of the diffusion model employed?

-  In sparse VAE's total objective, $ L_{Occ} $ (occupancy loss) and $\lambda_7$ are undefined, with no calculation/role explained .

- What is the parameter count of this model?

- In bridge diffusion SDE, $g(t)$ (diffusion coeff) and $f(x_t,t|\mathcal{S}_L)$ (drift term) lack definitions.

- What is the associated computational runtime (e.g., inference latency)?

---

> ### Author Response · Authors · 2025-11-22
>
> Thank you for taking the time to review our work and for providing thoughtful and helpful comments. We address each point in detail below.
>
> ### 1.The integration of geometric priors into the diffusion process to guide generation is a relatively established technique, as seen in works like DiffSplat and Hunyuan3D. Furthermore, leveraging an SMPL prior as a structural constraint has been demonstrated in several existing methods (e.g., SiFU, SiTH), which calls the novelty of this specific contribution into question.
>
> **Response**:
>  We clarify that we did not claim that integrating geometric priors into the diffusion process is the contribution of our paper. Instead, the novelty lies in how these priors are integrated into a unified latent formulation that enables single-stage generation. Specifically, our contribution centers on:
> (1) a sparse VAE that jointly compresses geometry and appearance of high-resolution 3D Gaussians into a single latent code;
> (2) a modality-unification module that transforms depth and SMPL into a consistent latent structural guidance; and
> (3) a bridge-latent diffusion model operating in this unified latent space for one-step reconstruction.
> This combination allows efficient joint modeling of geometry and appearance, which is not achieved in prior works such as DiffSplat, Hunyuan3D, SiFU, or SiTH.
>
> ### 2.The paper provides insufficient visualization results, making it difficult to assess the 3D consistency of the proposed method. Based on prior work in human avatar generation, a minimum of 15-20 video results are typically required to substantiate generalization claims. Comparative video analysis against other methods is also strongly recommended. The presented qualitative results are unconvincing. Specifically, Fig. 2 and Fig. 3 exhibit significant loss of detail in facial regions and clothing. Moreover, the model performs poorly in challenging scenarios, such as the loose-fitting dress depicted in the third row. This apparent qualitative failure contrasts sharply with the reported quantitative metrics (e.g., PSNR 29.91, SSIM 0.943), which are surprisingly high, approaching the performance ceiling typically associated with multi-view reconstruction.
>
> **Response**:
> We have uploaded **an additional video demo** to better demonstrate the 3D consistency of our method. We also updated the qualitative results by including more challenging cases (e.g., hard poses and loose clothing). Regarding the quantitative scores, PSNR is computed only within the valid foreground mask, which is consistent with prior works and equally applied to all baselines. As shown in Table 1, other recent methods, such as Human3Diffusion, also achieve similarly high values under this evaluation protocol.
>
> ### 3. A fundamental representational trade-off exists: mesh-based methods typically excel in geometric fidelity (e.g., sharp contours, fewer artifacts/floaters), whereas 3DGS representations generally offer superior texture modeling (e.g., facial details, complex clothing). Achieving state-of-the-art (SOTA) performance in both geometry and texture simultaneously within a single representation is non-trivial. It is recommended that the authors align their evaluation benchmark with established standards in existing literature.
>
> **Response**:
> For geometry evaluation, we follow the standard practice in single-image human reconstruction and report the three most widely adopted metrics—Chamfer Distance (CD), point-to-surface distance (P2S), and normal consistency—consistent with classic methods such as PIFu, ICON, and ECON. For appearance evaluation, we also use the widely established metrics PSNR, SSIM, and LPIPS. Therefore, our benchmark protocol is fully aligned with the evaluation standards commonly used in prior work.

---

> ### Author Response · Authors · 2025-11-23
>
> ### 4. The paper omits comparisons with several critical and highly relevant baselines in human avatar generation, notably HumanSplat [1], Human3Diffusion [2], and PSHuman [3]. Empirically, these human-centric models demonstrate superior texture fidelity and overall quality compared to general object generation models, making them essential points of comparison.
>
> **Response**:
> HumanSplat has not released complete training or inference code, which makes a direct and fair comparison infeasible. However, we have cited and discussed this work in the related-work section. For the remaining baselines, Human3Diffusion and PSHuman, we have added both quantitative and qualitative comparisons in the revised manuscript (See **Table 1, Fig.2, Fig.3 and Fig.4**).
>
> ### 5. Given that the method jointly utilizes multiple priors (DepthAnything V2, a diffusion prior, and an SMPL prior), a comprehensive ablation study is required. This study must meticulously dissect the contribution of each component to quantify its relative importance and impact on the final performance.
>
> **Response**:
> We have included ablation studies on the  priors used in our framework—depth estimation and SMPL estimation—and quantified their impact on performance in Section 4.2 of the manuscript.
>
> ### 6.Critical implementation details are omitted. For example, while citing relevant works [e.g., 4, 5], the authors do not specify the exact architecture of the diffusion model employed as a prior, its parameter count, or the computational runtime.
>
> **Response**:
> All critical implementation details, including the architecture, parameter count, and runtime, have been added to Appendix A.1 of the revised manuscript.
>
> ### 7. In sparse VAE's total objective,  $L_{occ}$(occupancy loss) and $\lambda_7$ are undefined, with no calculation/role explained .
>
> **Response**:
> We have given the definition of the occupancy loss, g(t) and f(x_t,t|S_L). ($\lambda_7$ is already defined **at Page 5 Line 244** in the original manuscript)

---

> ### Comment · Reviewer_gV1m · 2025-11-26
>
> I thank the authors for their detailed responses, the additional comparative experiments (specifically regarding `Human3Diffusion` and `PSHuman`), and the supplementary video materials. I have carefully reviewed the revised manuscript and the discussions with other reviewers. I appreciate the authors' effort in clarifying the implementation details and adding the necessary ablations regarding the Depth and SMPL priors. These additions have addressed my initial concerns regarding the completeness of the evaluation and technical reproducibility. `Consequently, I am willing to raise my score from 2 to 4`.
> And I recommend that the key clarifications and analyses from the rebuttalbe explicitly incorporated into the final version (A more detailed discussion).
>
> Reconstructing human avatars from a single image or sparse images is a challenging and significant problem. I commend the author's efforts and believe this is a promising research direction.  However, despite these improvements,  `I still believe the paper falls short of the high bar required for publication at ICLR at this stage` (from my view). The paper primarily compares against generative methods for general objects and leverages foundational vision models (e.g., Depth Anything). However, the work could be strengthened by a more thorough investigation and comparison of methods that explicitly incorporate human-specific priors, rather than merely incremental improvements upon prior approaches (e.g., HaP).
>
>  While the method is technically sound, the qualitative results (from the 360 videos and qualitative results in the main paper) indicate that there is still significant room for improvement compared to the very latest SOTA methods. From my perspective, the core contribution still leans more towards a combination of existing modules rather than a fundamental methodological breakthrough.
>
> For instance, `regarding the methodological synergy`, while the 'unified latent formulation' is the claimed novelty, the approach fundamentally relies on assembling established strong priors. The admission in the rebuttal that the model requires both priors to function suggests the core diffusion model lacks the robustness to hallucinate or correct details when those constraints are imperfect. Furthermore, observing the new video and comparison figures, there remains a visible gap in high-frequency texture details compared to methods like `PSHuman`. The current sparse VAE compression appears to be losing too much information, resulting in blurriness that limits practical utility. Finally, the reliance on SMPL priors acts as a double-edged sword; while it stabilizes the pose, it clearly limits the generation of loose clothing. To reach the next level, the method would need a mechanism to break free from the SMPL constraint in regions with high geometric uncertainty, rather than over-smoothing them.
>
>
> Once again, I thank the authors for their hard work and comprehensive response.

---

> ### Author Response · Authors · 2025-11-26
>
> Dear Reviewer **gV1m**
>
> We appreciate the reviewer for carefully reading both our manuscript and our previous responses, and for taking the time to update the score. We hope that the additional clarifications provided below can help address the remaining concerns and allow the reviewer to reconsider the score.
>
> ### **1. Clarification on novelty.**
>  We would like to clarify our position regarding the novelty of this work. In the field of human reconstruction, fundamental methodological breakthroughs are **rare** because the task is inherently a **system-level problem**.
>
> For example, **PIFu (ICCV 2019)** was the first to apply implicit functions to human reconstruction, but implicit functions themselves were not invented in that work.
> Similarly, **PSHuman (CVPR 2025)** relies on multi-view diffusion and continuous remeshing, yet neither technique is original to PSHuman.
>
> These methods advance the field primarily by proposing new frameworks that integrate existing components in novel and effective ways. Such **framework-level breakthroughs** are the dominant and most impactful form of innovation in human reconstruction.
>
> In the same spirit, our goal is not to introduce a new primitive algorithm, but to design a new framework that enables a single-step diffusion model to jointly learn geometry and appearance—something no prior method is structurally capable of achieving. Thus, the contribution of our work also lies at the framework level, which is both appropriate and meaningful for this domain.
>
> ### **2. Comparison with existing methods.**
> Quantitatively, our method outperforms PSHuman and other recent approaches, as shown in Table 1. Qualitatively, we acknowledge that the VAE compression may sacrifice certain fine-grained details compared to PSHuman. However, our method shows significantly better global structure and pose consistency, which are often more critical in human reconstruction. High-frequency details lose practical value if the underlying shape or pose is implausible. Therefore, this work prioritizes structural consistency over purely local refinement.
>
> We have also conducted extensive comparisons with several SMPL-based methods (GTA, SIFU, SITH, IDOL, MultiGO) in the **original manuscript**, and our approach consistently achieves better overall quality.
>
> ### **3. Role of SMPL.**
> We also clarify the role of SMPL in our framework. SMPL provides a strong prior for global body structure and pose, while depth information supplies rich geometric cues for clothing. These two sources of information are complementary: the SMPL prior ensures coherent global structure, and the depth map constrains local garment geometry. Therefore, SMPL does not limit the reconstruction of loose clothing; instead, it improves stability in challenging poses. As shown in **the last row of Fig. 5**, our method performs robustly across various loose-clothing examples.
>
> A concrete example appears in the **first case of Fig. 4**. The subject performs **a large forward stride**, where the leg distance should **naturally be wide**. PSHuman incorrectly reconstructs the legs as being very close together, leading to a pose that appears acceptable from the front but clearly implausible from side and back views. Our method preserves the correct leg spacing and overall pose, thanks to the SMPL structural prior.
>
> Additionally, in the **detailed comparisons in Fig. A2**, PSHuman shows noticeable tearing along clothing boundaries and unstable facial details. Our method produces more consistent and reliable results in these regions.
>
> We agree with the reviewers' suggestion and will explore more natural and elegant ways of integrating SMPL pose information in future work.

---

> > ### Comment · Reviewer_gV1m · 2025-11-26
> >
> > I have read the authors' rebuttal regarding the novelty and the comparison with PSHuman. I have carefully considered the arguments concerning "framework-level breakthroughs" and the trade-off between structural consistency and texture fidelity. While I appreciate the authors' perspective and the effort put into the rebuttal, I must maintain my current assessment. My decision is grounded in the following core concerns that remain unresolved:
> >
> > 1. While the system integration is sound, but it feels largely like a combination of existing tools (DepthAnything, SMPL) to control the diffusion process. Which makes this work more like an engineering trade-off (gaining stability at the cost of fidelity) rather than a fundamental methodological breakthrough. The results do not demonstrate a clear leap over existing paradigms to justify the compromised texture quality.
> >
> > 2. The blurry results are my another concerns. The authors admit that the Sparse VAE compression hurts fine details. For avatar generation, realistic texture is critical. The results are still visibly blurry, suggesting a hard ceiling on quality inherent to this architecture. For high-fidelity avatars, this lack of realism is a dealbreaker when compared to SOTA.
> >
> > Overall, the work is solid, but the performance limitations currently prevent it from meeting the bar for acceptance. I thank the authors' rebuttal, I remain open to further discussion with the authors and the other reviewers / ACs.

---

> > > ### Author Response · Authors · 2025-11-26
> > >
> > > We thank the reviewer for the careful consideration and constructive comments.
> > >
> > > We would like to clarify that the contribution of our work is not merely a combination of existing tools (DepthAnything, SMPL, diffusion). Instead, we propose a new system-level framework that enables single-step 3D diffusion to jointly model both geometry and appearance—a capability that no prior method provides. It is important to note that almost all works in human reconstruction are system-level research projects that integrate multiple components; low-level algorithmic breakthroughs are rare in this field. Meanwhile, simply combining these components is insufficient for effective human reconstruction. The key lies in designing the system such that the different components work collaboratively and complement each other, which requires careful architectural design and thoughtful integration.
> > >
> > > Our experiments demonstrate that our method quantitatively outperforms all compared methods， as shown in Table 1. Qualitatively, while some fine-grained details may be less sharp than PSHuman, our results consistently preserve global body structure, pose plausibility, facial geometry, and loose clothing, as shown in the updated visualizations in Figure 4, Figure 5. PSHuman is based entirely on a 2D diffusion paradigm, whereas our approach operates directly in 3D, representing a meaningful advancement in 3D human reconstruction.
> > >
> > > We acknowledge that no method can perfectly reconstruct a human from a single image, and occasional limitations in details exist across all current approaches, including both PSHuman and ours. However, these limitations do not undermine the framework-level contribution of our work. We believe our method represents a significant step forward in the 3D diffusion paradigm for human reconstruction, and we hope the reviewer may reconsider the assessment in light of this perspective.

---

### Official Review · Reviewer_dZF7 · 2025-10-30

**Soundness:** 2
**Presentation:** 1
**Contribution:** 2
**Rating:** 2
**Confidence:** 5

**Summary:**

This paper proposes JGA-LBD to address the challenge of reconstructing both human geometry and appearance from a single RGB image. The core contribution is to formulate this as a "bridge diffusion" task within a "unified latent space," which jointly encodes both geometry and appearance. The method takes depth and SMPL priors, unifies them into a 3D Gaussian (3DGS) representation, and then compresses this into a compact latent space. A bridge diffusion model is then used to reconstruct the complete 3D human from this latent code. Experimental results demonstrate that this approach yields improvement over 3DGS-based methods.

**Strengths:**

1. The model handles a case involving a child, which is typically a difficult body shape to reconstruct.
2. Although the method is based on an SMPL prior, the final visual results appear to overcome some of SMPL's common limitations and achieve a good-quality outcome.

**Weaknesses:**

### **Major**

1. **Methodological Clarity:** The core methodology is not explained with sufficient clarity and is difficult to follow. Several key components are either ambiguous or inadequately described.
    - The illustration in Fig. 1 is confusing. The "Depth" and "SMPL" inputs are depicted with color, which is misleading. Furthermore, visualizing these inputs in a 3DGS style implies a conversion has already occurred.
    - **Input Conversion:** The paper fails to explain the process of converting Depth and SMPL data into the 3DGS-style representations shown. If this involves intermediate steps (e.g., depth back-projection to a colored point cloud, or SMPL vertex color gathering), these intermediate modalities are incorrectly labeled. For clarity, I strongly suggest renaming them to reflect their true nature, such as "depth-derived 3D points" or "SMPL-derived mesh."
    - **3DGS and Voxel Grids:** The association between the 3DGS representation and voxel grids (L158) is not explained. The authors should clarify if a voxelization process is used and detail its role in the pipeline.
2. **VAE Training Process:** The training procedure for the compression VAE is a critical, unexplained step. L215 states that "ground truth 3DGS" is used for training, but the cited datasets (Thuman2.1, 2k2k) do not natively provide data in this format; L309 mentions that 1600 scans were used to "prepare" these attributes, but the preparation process itself is not detailed. This omission makes the method difficult to reproduce and its foundation unclear.
3. **Missing Ablation Studies:** The design choice of integrating both depth and SMPL priors is not empirically justified.
    - Given that SMPL models inherently contain 3D shape and (implicit) depth information, the necessity of an additional 2D depth map modality is not obvious.
    - The paper would be significantly strengthened by an ablation study that evaluates the performance of the method using only SMPL priors, without depth (and using only depth priors with SMPL). Both quantitative and qualitative results should be provided for this comparison.
4. **Insufficient Experimental Comparison:** The experimental validation is lacking. The paper only compares against other 3DGS-based methods. There is a severe lack of comparison against mainstream methods that generate textured meshes (e.g., ECON, SiTH, GTA, PSHuman). Without this context, the claimed superiority of the method is unsubstantiated. (Please refer to the qualitative comparisons in the paper of IDOL.)
5. **Notation and Formalism:** The paper's notation is imprecise. Key variable dimensions (e.g., the latent dimension's shape, whether 1D or 3D) are not specified. Distinct notations should be introduced for the initial inputs (Depth, SMPL) versus their processed derivatives (e.g., depth-derived points, SMPL-derived mesh) and their corresponding latent representations.

### **Minor**

1. **Missing Related Work**: The paper omits a large and relevant category of optimization-based (SDS) methods, which often have better generalization, including:
    - TeCH: Text-guided Reconstruction of Lifelike Clothed Humans
    - Human-SGD: Single-Image 3D Human Digitization with Shape-Guided Diffusion
    - GeneMAN: Generalizable Single-Image 3D Human Reconstruction from Multi-Source Human Data
2. **Minor Grammatical Error**: L223 contains a grammar mistake ("enables generate" instead of "enables generating").

**Questions:**

1. Why does a method based on an SMPL prior (which can introduces pose errors) ultimately achieve such good results? Is this improvement come from depth regularization?

---

> ### Author Response · Authors · 2025-11-22
>
> We sincerely appreciate your time and the constructive feedback on our manuscript. Our detailed, point-by-point responses are presented below.
>
> ### Major1. Methodological Clarity:The illustration in Fig. 1 is confusing. The "Depth" and "SMPL" inputs are depicted with color, which is misleading. Furthermore, visualizing these inputs in a 3DGS style implies a conversion has already occurred.
>
> **Response**:
> Sorry for the confused illustration. We have updated Fig.1 and renamed some components. The input of the sparse VAE is human 3D Gaussian attributes, depth-derived 3D Gaussian attributes and SMPL-derived 3D Gaussian attributes. The conversion has already occurred before training the sparse VAE.
>
> ### Major1. Input Conversion: The paper fails to explain the process of converting Depth and SMPL data into the 3DGS-style representations shown. If this involves intermediate steps (e.g., depth back-projection to a colored point cloud, or SMPL vertex color gathering), these intermediate modalities are incorrectly labeled. For clarity, I strongly suggest renaming them to reflect their true nature, such as "depth-derived 3D points" or "SMPL-derived mesh."
>
> **Response**:
> We clarify that the conversion process is explicitly described in Section 3.1, and no intermediate modalities are incorrectly labeled. The depth map is back-projected into a partial point cloud, which is clearly named as such (**Page 4, L190 and L196**). For the SMPL branch, we directly use the standard SMPL mesh representation, as is common in prior work (e.g., ICON, ECON). We also specify that SMPL vertices are converted into 3D Gaussian attributes (**Page 4, L202**).
>
> ### Major1.  3DGS and Voxel Grids: The association between the 3DGS representation and voxel grids (L158) is not explained. The authors should clarify if a voxelization process is used and detail its role in the pipeline.
>
> **Response**:
> We **indeed** mention that the 3D Gaussian positions are represented on voxel grids, as stated on **Page 3, Line 158** of our original submission. You may have **overlooked** this point. This voxelization step simply organizes the 3D points into sparse grids so they can be processed by our sparse 3D CNNs.
>
> ### Major2. VAE Training Process: The training procedure for the compression VAE is a critical, unexplained step. L215 states that "ground truth 3DGS" is used for training, but the cited datasets (Thuman2.1, 2k2k) do not natively provide data in this format; L309 mentions that 1600 scans were used to "prepare" these attributes, but the preparation process itself is not detailed. This omission makes the method difficult to reproduce and its foundation unclear.
>
> **Response**:
>  We clarify that we have already mentioned the preparation strategy at **Page 3 Line 161** in the original manuscript.
>
> ### Major3. Missing Ablation Studies:The design choice of integrating both depth and SMPL priors is not empirically justified. Given that SMPL models inherently contain 3D shape and (implicit) depth information, the necessity of an additional 2D depth map modality is not obvious. The paper would be significantly strengthened by an ablation study that evaluates the performance of the method using only SMPL priors, without depth (and using only depth priors with SMPL). Both quantitative and qualitative results should be provided for this comparison.
>
> **Response**:
> Depth and SMPL provide complementary information in our framework. Depth supplies visible fine-grained details such as clothing and hair contours, which SMPL cannot represent. However, Depth is limited to only the visible regions and cannot recover the occluded body structure or pose, which SMPL provides. Consequently, relying on either modality alone leads to an incomplete supervision signal, causing the diffusion-generated latent codes to fail during decoding (the Minkowski engine core will crash and output nothing.
> Integrating SMPL with additional geometric cues is a standard practice in prior works such as ICON, ECON, SiTH, GTA, and SiFu. Suggested by Reviewer NQHJ, we have provided an additional ablation study of these two modules in Section 4.2.
>
> ### Major4. Insufficient Experimental Comparison: The experimental validation is lacking. The paper only compares against other 3DGS-based methods. There is a severe lack of comparison against mainstream methods that generate textured meshes (e.g., ECON, SiTH, GTA, PSHuman). Without this context, the claimed superiority of the method is unsubstantiated. (Please refer to the qualitative comparisons in the paper of IDOL.)
>
> **Response**:
> First, ECON cannot generate textured meshes. Second, **we clarify that we have already compared SiTH and GTA in Table. 1 in the original manuscript.** We have compared our methods with the three most recent CVPR 25 papers both quantitatively and qualitatively.  And we have provided a comparison with PSHuman in Table. 1 and Figure. 4.

---

> ### Author Response · Authors · 2025-11-23
>
> ### Major5.  Notation and Formalism: The paper's notation is imprecise. Key variable dimensions (e.g., the latent dimension's shape, whether 1D or 3D) are not specified. Distinct notations should be introduced for the initial inputs (Depth, SMPL) versus their processed derivatives (e.g., depth-derived points, SMPL-derived mesh) and their corresponding latent representations.
>
> **Response**:
> We clarify that the latent dimension is clearly presented at **Page 6 Line 290-291**.  The definition of each notation is clearly defined at **Page 4 Line 215, Page 5 Line 244-245**.
>
> ### Minor1. Missing Related Work: The paper omits a large and relevant category of optimization-based (SDS) methods, which often have better generalization.
>
> **Response**:
> We have cited and discussed these papers in this ICLR manuscript.
>
> ### Minor2. Minor Grammatical Error: L223 contains a grammar mistake ("enables generate" instead of "enables generating").
>
> **Response**:
> We have corrected the grammatical mistake.
>
> ### Q1. Why does a method based on an SMPL prior (which can introduces pose errors) ultimately achieve such good results? Is this improvement come from depth regularization?
>
> **Response**:
> We refine the initial SMPL pose following the HaP [1] strategy, which substantially reduces pose errors and leads to improved final performance. Depth further provides strong geometric constraints on visible regions, complementing the SMPL prior and enhancing the reconstruction quality.
> [1] "Human as points: Explicit point-based 3d human reconstruction from single-view rgb images." IEEE Transactions on Pattern Analysis and Machine Intelligence (2025).

---

### Official Review · Reviewer_NQHJ · 2025-11-01

**Soundness:** 2
**Presentation:** 3
**Contribution:** 2
**Rating:** 4
**Confidence:** 5

**Summary:**

The paper proposes JGA-LBD, a method that reconstructs both 3D geometry and appearance of humans from single RGB images using bridge diffusion in a unified latent space. The key innovation is compressing heterogeneous modalities (depth maps and SMPL models) into 3D Gaussian representations, then learning to generate complete human models via bridge diffusion.

**Strengths:**

1.This jointly models smpl, geometry and appearance in a single latent space, ensuring better consistency.

2.Outperforms state-of-the-art on both 2K2K and CustomHuman benchmarks across all metrics (geometry: CD, P2S, Normal; appearance: PSNR, SSIM, LPIPS)

**Weaknesses:**

1.Sparse convolutions with Minkowski Engine and Bridge diffusion are standard established techniques. The main contribution is primarily engineering/combination rather than fundamental innovation.

2. VAE reconstructed results seems blurry, how this vae is compared with other formulations such as Trellis vae?

3.The in-the-wild examples show relatively standard poses. Challenging cases aren't thoroughly evaluated.

4.Looth clothes cases are not well presented.

5.The "unified latent space" claim is somewhat oversold - depth and SMPL are converted to 3DGS format but remain distinct modalities.

6.Missing references: TeCH: Text-guided Reconstruction of Lifelike Clothed Humans. In 3DV 2024
MagicMan: Generative Novel View Synthesis of Humans with 3D-Aware Diffusion and Iterative Refinement. In Arxiv 2024
Generalizable Human Gaussians from Single-View Image. In ICLR 2025
Humangif: Single-view human diffusion with generative prior. In Arxiv 2025
WonderHuman: Hallucinating Unseen Parts in Dynamic 3D Human Reconstruction. In Arxiv 2025

**Questions:**

How does the method perform when upstream depth/SMPL estimation fails? The paper doesn't analyze failure modes or provide robustness analysis when Depth Anything V2 or PIXIE produce incorrect predictions.

What is the actual computational cost? Training time, inference time, and memory requirements are not reported. With 200k iterations for VAE and 100k for diffusion on 4× A6000 GPUs, this seems computationally expensive.

Why not learn depth and SMPL prediction end-to-end? The modular pipeline with frozen pre-trained models may be suboptimal compared to joint training.

---

> ### Author Response · Authors · 2025-11-22
>
> Thank you for your valuable time and for the insightful comments on our paper. Our point-by-point responses are provided below.
>
>
> ### 1. Sparse convolutions with Minkowski Engine and Bridge diffusion are standard established techniques. The main contribution is primarily engineering/combination rather than fundamental innovation.
>
> **Response**:
> Reconstructing 3D digital humans is a complex task that cannot be solved by a single simple technique. It is a system-level problem that naturally requires leveraging multiple existing components. For this reason, most 3D human reconstruction methods innovate by proposing new *frameworks/pipelines*, rather than by reinventing all underlying techniques.
> For example, PIFu did not introduce implicit functions, and the paper ***Generalizable Human Gaussians from Single-View Image*** suggested by the reviewer also relies on existing modules such as 3DGS, SMPL-X and ControlNet.
> This does not reduce their contributions. These methods are considered innovative because their frameworks effectively address important challenges and help move the field forward.
> Our work follows the same principle. We propose a new framework that unifies geometry and appearance into a single latent representation for joint learning—something that previous methods could not achieve. This unified framework is therefore our main and most important contribution
>
> ### 2. VAE reconstructed results seems blurry, how this vae is compared with other formulations such as Trellis vae?
>
> **Response**:
> Although Trellis can also generate a 3D object from a single image, the core idea behind our method is fundamentally different. Trellis appears to use a VAE latent representation, but its VAE does **not** encode geometry and appearance together, Trellis  needs an additional diffusion step to generate the sparse structure to provide geometry information. In contrast, our VAE jointly compresses both geometry and appearance into a single latent, which allows our diffusion stage to operate in only one step.
> Because our input and output settings are also different, the two VAEs are not aligned in design, so it is not meaningful to directly compare their VAE performance.
> Instead, we compare the final reconstruction results in the paper. The results show that Trellis often produces inconsistent geometry and appearance because its VAE cannot represent them jointly. Our unified latent representation avoids this issue by learning geometry and appearance together. Therefore, the key innovation of our work is this unified latent space, which is more effective than the separated design used in Trellis.
>
> ### 3&4. The in-the-wild examples show relatively standard poses. Challenging cases aren't thoroughly evaluated. Looth clothes cases are not well presented.
>
> **Response**:
> We have provided more challenging poses and loose clothing in Fig. 4, Fig.5 and the video demo.
>
> ### 5. The "unified latent space" claim is somewhat oversold - depth and SMPL are converted to 3DGS format but remain distinct modalities.
>
> **Response**:
> The depth map is converted to a partial point cloud, and then converted to 3DGS format. And the vertices (which can also be considered as point cloud, the faces of the SMPL are not used in our paper.) of SMPL are also converted to 3DGS format. Since they are all in 3DGS format, they are in the same modality.
>
> ### 6. Missing references
>
> **Response**:
> We have cited and discussed these papers in this ICLR manuscript.

---

> ### Author Response · Authors · 2025-11-23
>
> ### Q1. How does the method perform when upstream depth/SMPL estimation fails? The paper doesn't analyze failure modes or provide robustness analysis when Depth Anything V2 or PIXIE produce incorrect predictions.
>
> **Response**:
> In practice, upstream modules rarely “fail” catastrophically; instead, the quality of depth and SMPL estimation varies, and our system always produces valid depth and SMPL predictions. We have updated the ablations to quantify this effect: Depth Anything V2 (RMSE 0.014 on 2K2K) leads to better downstream performance than EcoDepth (RMSE 0.016 on 2K2K), and similarly different SMPL regressors (PIXIE vs. PyMAF) yield slightly different but comparable results (Table 2). Since 2K2K does not provide ground-truth SMPL meshes (the authors have not yet responded), a full quantitative SMPL comparison is not possible, but we will update our manuscript once the data becomes available.
>
> ### Q2. What is the actual computational cost? Training time, inference time, and memory requirements are not reported. With 200k iterations for VAE and 100k for diffusion on 4× A6000 GPUs, this seems computationally expensive.
>
> **Response**:
> We have provided the implementation details in the manuscript section Appendix. And the computation is not expensive, for example, your suggested paper MagicMan needs 8 $\times$ A100 GPUs, which needs much more computational resources than our method.
>
> ### Q3. Why not learn depth and SMPL prediction end-to-end? The modular pipeline with frozen pre-trained models may be suboptimal compared to joint training.
>
> **Response**:
> End-to-end training of depth and SMPL is theoretically appealing but remains extremely challenging in practice. Current methods—including all your suggested paper—also rely on multi-stage or frozen components because joint optimization across dense depth and low-dimensional SMPL parameters is highly unstable and lacks unified supervision. We consider fully end-to-end learning an important future direction.

---

### Author Response · Authors · 2025-11-23

We thank the AC and reviewers for their time and for the constructive feedback that helped us improve the quality of our paper. We have revised the manuscript according to the reviewers’ comments. The main updates are as follows:

1. Added missing citations.

2. Included comparisons with two additional methods—Human3Diffusion and PSHuman—and added more challenging-pose and loose-clothing examples (updated in Fig. 2, Fig. 3, Fig. 4, and Fig. 5).

3. Uploaded a **video demo** containing comparisons with other methods and additional in-the-wild results.

4. Added an ablation study on the depth and SMPL modules.

5. Updated the detailed network architecture as well as training and inference times.

We kindly ask the reviewers to consider the revised manuscript, the point-by-point response, and the new video demo when reassessing our work.

---

### Author Response · Authors · 2025-11-25
**Looking forward to your further assessment**

Dear **Reviewers**,

Thank you for taking the time to review our manuscript and for your valuable feedback. We have carefully addressed all the comments and concerns raised, as reflected in our detailed responses and the revised manuscript and supplementary material.

We sincerely appreciate your efforts and look forward to your further assessment.

Best regards,

The Authors

---

### Note · Authors · 2025-12-31

I have read and agree with the venue's withdrawal policy on behalf of myself and my co-authors.